# Effect of Differentiated Injection Ratio, Gas Flow Rate, and Slag Thickness on Mixing Time and Open Eye Area in Gas-Stirred Ladle Assisted by Physical Modeling

**Luis E. Jardón-Pérez** [1], **Daniel R. González-Morales** [1], **Gerardo Trápaga** [2], **Carlos González-Rivera** [1] and **Marco A. Ramírez-Argáez** [1,*]

[1] Facultad de Química, Universidad Nacional Autónoma de México, Avenida Universidad 3000, Cd. Universitaria, Coyoacán, Cd. de México 04510, Mexico; dregwar@gmail.com (L.E.J.-P.); daniel_ricardo21@hotmail.com (D.R.G.-M.); carlosgr@unam.mx (C.G.-R.)

[2] CIATEQ, Avenida del Retablo no. 150 col. Constituyentes FOVISSSTE, C.P. 76150 Querétaro, Qro., Mexico; gerardo.trapaga@ciateq.mx

\* Correspondence: marco.ramirez@unam.mx; Tel.: +52-5556-22-5246

**Abstract:** In this work, the effects of equal (50%/50%) or differentiated (75%/25%) gas flow ratio, gas flow rate, and slag thickness on mixing time and open eye area were studied in a physical model of a gas stirred ladle with dual plugs separated by an angle of 180°. The effect of the variables under study was determined using a two-level factorial design. Particle image velocimetry (PIV) was used to establish, through the analysis of the flow patterns and turbulence kinetic energy contours, the effect of the studied variables on the hydrodynamics of the system. Results revealed that differentiated injection ratio significantly changes the flow structure and greatly influences the behavior of the system regarding mixing time and open eye area. The Pareto front of the optimized results on both mixing time and open eye area was obtained through a multi-objective optimization using a genetic algorithm (NSGA-II). The results are conclusive in that the ladle must be operated using differentiated flow ratio for optimal performance.

**Keywords:** gas-stirred ladle; physical modeling; optimization; particle image velocimetry; steelmaking

---

## 1. Introduction

The ladle furnace plays a key role in the production of high-quality steel, because in this reactor several processes are carried out, such as desulfurization, deoxidation, chemical and temperature control, and elimination or modification of inclusions [1]. Most of the ladle objectives can be improved by achieving an efficient mixing of the liquid, and mixing time has been used extensively as an efficiency criterion of the process.

Accordingly, there has been a great interest to understand the effect of different process parameters on the mixing time in ladles, [2–4] including the gas flow rate, the number of plugs, the radial and angular position of the plugs in systems with multiple gas injection points, the presence or absence of slag, the thickness of the slag layer, and the diameter of the plugs, among other variables. The gas flow rate is by far the variable affecting the most the mixing time and it is well known that the larger the gas flow rate the shorter is the mixing time obtained in ladles. Regarding the number of plugs and their positions, Conejo et al. [5] reported the best mixing conditions with one plug located at 2/3 R, whereas González-Bernal et al. [6] reported the best injection point with one nozzle located at 3/4 R. In a recent review on ladle metallurgy, Liu et al. [4] summarize the most relevant contributions made

in the study of mixing time in the last three decades. In short, recommended plug configurations for the minimization of mixing time in the case of single-plug injection involve off-centered locations of the plugs ranging from 0.5 to 0.67 R, and in the case of dual injection, a separation angle of 180° is recommended, whereas the optimum location of the plugs is again at the mid-radius position between the center and the lateral wall of the ladle. All the studies presented in that review measured the local mixing time of a solute using conductivity and pH probes.

Recently, the authors of the present work found that mixing times measured by the planar laser-induced fluorescence technique (PLIF) are similar but more accurate than those obtained under optimal conditions by pH probes. Besides, mixing time measured through the PLIF technique does not depend on the location of the measurement instrument, as is the case with the pH probe method [7].

The slag layer plays a decisive role in the refinement of the steel in the ladle because this phase allows refining operations such as desulphurization and deoxidation, among other slag-metal exchanges, and also avoids metal exposure to the atmosphere, thus preventing oxidation of the metal and heat losses. As the ladle is a system agitated by the injection of inert gas, the rise of bubbles breaks up the slag layer exposing a certain area of the liquid metal to the atmosphere; this phenomenon is often called open eye area. Harmful phenomena that reduce the steel quality occur in the open eye such as reoxidation, nitrogen pickup, and possible slag entrapment in the steel bath. During ladle operation, it is well-known that low mixing times are easily obtained using violent gas stirring; however, large gas flow rates produce big eye areas. Therefore, the process aims to get short mixing times and at the same time small eye areas [3,4]. For one plug and centric gas injection, several studies on the effect of different process variables on the open eye area have been carried out. In general, the effect of increasing the gas flow rate is to increase the eye area, and the increase of the slag thickness decreases that area. The effect of gas flow rate, bath height, slag thickness, and the substance used to simulate the phase corresponding to the slag on the open eye area were studied by Peranandhanthan and Mazumdar [8] in a physical model. They found that the density and viscosity of the upper liquid phase can influence considerably the slag eye area. Lee and Yi [9] found that the open eye area is proportional to the square root of the gas flow rate and inversely proportional to the square root of the slag thickness. Lv et al. [10] analyzed, in a physical model, the effect of slag thickness, gas flow rate, and liquid height in the model, in addition to considering the scale of the model and the substance that simulates the molten steel on the eye area. They found that not only the gas flow rate, the liquid height, and the thickness of the slag layer had strong effect on the slag eye size, but also the transversal area of the vessel had great influence on that size. In all of the above-mentioned studies [8–10], correlations calculating the eye area as a function of operational variables are reported. Furthermore, the empirical equations for open eye [4,9–11] have been modified on the basis of parameters such as the density ratios of the bulk and slag phases, the Froude number, and the Reynolds number.

There are also studies that simultaneously consider the mixing time and the eye area. Liu et al. [12] used physical modeling to study the effect of the gas flow rate, the number of plugs, and the angle between plugs on mixing time and open eye area, and recommended a dual injection located at 2/3 R with a separation angle of 180°, which results in low mixing times and a stable metal-slag interface. Amaro-Villeda et al. [13] used physical modeling to study the effect of the number of plugs, their radial position, the gas flow rate, and the properties of the upper liquid phase on mixing time and open eye area. They reached the shortest mixing time with a single plug located at R/2. Recently, Liu et al. [14] studied the effect of the single or dual injection modes, the plug location, the angle between plugs, the gas flow rate, and the slag thickness of the upper phase on mixing time and open eye formation. For one plug, they found that the slag eye area increases only by a small proportion when the porous plug location moves from the center toward the wall of the vessel, whereas for two plugs the slag eye area increases significantly when the angle between plugs increases from 60° to 180°.

Open eye and mixing time are not the only parameters to be optimized in the ladle. Geng et al. [15] studied the non-metallic inclusion removal rate from the steel to the slag phase through a mathematical model. They found the best configuration for inclusion removal are 3 plugs at R/2, two of them

separated 180° and the last one 90° from the former two plugs, since the inclusions are dragged by the ascending steel motion towards the slag but then, the inclusions have enough residence time in contact with the slag in a horizontal motion to improve the removal rate.

Although efforts have been made to optimize the operation of the ladle, in most cases only the minimization of mixing time is considered and the optimization is not performed using formal algorithms [16–19]. Recently, Mazumdar et al. [11] and Jardón-Pérez et al. [20] performed optimizations of mixing time and slag eye area in physical models of a ladle using formal algorithms. In both works, optimization was done with the aim of finding the conditions that simultaneously minimize mixing time and the open eye area by setting properly a multi-objective and multi-variable optimization problem. In both works, Pareto fronts were obtained showing that the optimum solutions are located between two extreme scenarios: One, which minimizes mixing time but produces big eye areas, and the other showing the smallest eye area but long mixing times. Mazumdar et al. [11] identified a balanced compromise between mixing time and slag eye area in the Pareto front by pointing out a region in the front where the slope of the curve of the slag eye area versus the mixing time changes rapidly. They suggested controlling the process by varying the height of the bath, the slag thickness, and the gas flow rate for various injection points. Jardón-Pérez et al. [20] have performed, additionally, a similarity analysis among Pareto optimal solutions and Pareto characterization to identify the operational rules, which include an optimal injection position with two plugs located at 180° and 4/5 R, and variation of gas flow rate and slag thickness, in order to operate in three different ranges of mixing time.

One of the least studied variables in the case of dual gas injection is the use of different gas flow rates in each plug (a gas blowing ratio different to 1:1). Chattopadhyay et al. [16] reported that the effect of using a differentiated flow ratio is not conclusive, because it presents an improvement in mixing in some injection positions but a detriment in others. Liu et al. [12] reported a small improvement in mixing time when performing a differentiated gas blowing with the same operating conditions. More recently, Haiyan et al. [21] found that the gas blowing ratio influences the mixing on the ladle and that the use of different gas flow rates can significantly decrease mixing time, compared with the mixing time reached when using the same flow rates. They performed the validation of a CFD model through the experimental results, which were better explained through the analysis of the model predictions. In the case of differentiated flows, the mathematical simulation of the flow fields shows that the eye of the loops of the two plumes are not located at the same height. This is due to the difference in the gas flow rates injected at each plug (i.e., the strong plume forms a larger circulation loop stirring most of the ladle, whereas the weak plume forms a smaller circulation). Then, the interaction and collision between the loops is weaker than that for equal flows, allowing the stirring energy to be used more efficiently, which in turn could explain the observed decrease in mixing time.

In the present work the effect of gas blowing mode (equal or differentiated flow ratios), gas flow rate, and slag thickness on mixing time and open eye area are studied in a physical model of a ladle with dual plugs in order to determine correlations for open eye area and mixing time by statistical analysis of the results, based on Particle Image Velocimetry (PIV) measurements of the hydrodynamics and Planar Laser-Induced Fluorescence (PLIF) measurements for mixing time. Furthermore, the optimization of these results using a multi-objective genetic algorithm (NSGA-II) was performed to determine whether the use of differentiated flow ratio in gas-stirred ladles can be advantageous in simultaneously reducing both mixing times and open eye areas in comparison with the conventional equal blowing mode in dual injection systems.

## 2. Materials and Methods

A 1/17th physical model (an acrylic cylinder of 0.185 m diameter, 0.214 m height, and 0.17 m liquid level) of a 200 ton prototype employing distilled water, oil ($\mu = 0.19\ \mathrm{kg\ m^{-1}\ s^{-1}}$, $\rho = 890\ \mathrm{kg\ m^{-3}}$), and air to simulate steel, slag, and argon, respectively, and satisfying geometric, kinematic, and similarity criteria [13] was used to perform a process analysis by varying the gas injection ratio ($Ra$), the gas flow rate ($Q$), and the oil thickness ($hs$), and measuring mixing time and open eye area. Oil thickness in

percentage value is equivalent to its volume percentage, and since the system is a cylinder this could be calculated through the oil thickness (*hs*) divided by the total bath thickness (water thickness plus oil thickness, i.e., Hb + *hs*) times 100. Physical model meets geometric similarity with a scale factor 1/17 of an industrial ladle furnace. The kinematic similarity was met since water at room temperature and liquid steel at 1600 °C kinematic viscosities are the same. The prototype, an industrial ladle furnace operating at $Q_{fs}$ gas flow rate, was scaled through Equation (1) proposed by Mazumdar et al. [22] to the gas flow rate in a physical model ($Q_{mod}$) with a geometric scale factor $\lambda$ = 1/17 to meet the dynamic similarity in a gas stirred ladle where agitation is dominated by the modified Froude number. The gas flow rates for the physical model were between 1.54 and 2.22 L/min and in practice these were set with a couple of Cole Palmer flow meters.

$$\frac{Q_{mod}}{Q_{fs}} = \lambda^{2.5} \tag{1}$$

Unfortunately, it is difficult to satisfy simultaneously the dynamic and kinematic similarity for the oil phase representing slag, but it is better to present a top layer of liquid that dissipates stirring energy than not presenting such a phase. The use of such a small vessel helps the visualization of the tracer evolution through the PLIF technique, which is very beneficial in the quantitative study of mixing of a solute gaining basic insight of these phenomena, and, consequently, it is worth using this small-scaled system despite the possible wall effect overprediction.

A full-factorial experimental design at two levels with the mentioned three variables was performed by duplicate and the $2^3$ experiments are presented in Table 1, where the high and low values of the variables were set.

**Table 1.** Experimental design with high and low values of the three variables analyzed.

| Experiment | Oil Thickness (*hs*) (%) | Gas Flow Rate (*Q*) (L min$^{-1}$) | Injection Ratio (*Ra*) (%/%) |
|:---:|:---:|:---:|:---:|
| a | 3 | 1.54 | 50/50 |
| b | 3 | 2.22 | 50/50 |
| c | 3 | 1.54 | 75/25 |
| d | 3 | 2.22 | 75/25 |
| e | 5 | 1.54 | 50/50 |
| f | 5 | 2.22 | 50/50 |
| g | 5 | 1.54 | 75/25 |
| h | 5 | 2.22 | 75/25 |

Figure 1a shows the setup of the particle image velocimetry (PIV) technique used to determine mean liquid velocities and turbulent kinetic energy at a longitudinal plane, including the nozzles and the center of the ladle as shown schematically in Figure 1b. The PIV technique uses a laser Litron® LDY302 (Litron Lasers Ltd., Rugby, UK) at 45% of its power (15 mJ) and 527 nm wavelength, synchronized to a SpeedSense® M320 (Dantec Dynamics A/S, Skovlunde, Denmark) camera operating in single frame mode at 450 Hz, to measure the flow pattern and turbulence in the physical model of a gas stirred ladle using a dual gas injection system controlled by two flowmeters Cole Palmer® (Cole-Parmer, Vermon Hills, IL, USA), with nozzles located at 4/5 of the ladle radius (optimum conditions according to Jardón et al. [19]). The model was seeded with polyamide spherical particles of 5 µm and the 1351 photographs taken were analyzed with the aid of the software DynamicStudio© 2015a (Dantec Dynamics A/S, Skovlunde, Denmark) to perform operations of calibration, image masking, cross-correlation, average filter, and vector statistics on raw images, divided in 32 × 32 pixel interrogation areas, of the plane of measurement (Plane 1 in Figure 1b) to obtain the velocity vector and turbulent kinetic energy (*k*) contour plots presented in this work. Equation (2) is the employed

method for the computation of *k* in this work. This equation assumes pseudo-isotropic turbulence in the perpendicular direction to the plane of measurement for 2D measurements by PIV [23].

$$k = \frac{3}{4}\left(\overline{u'^2} + \overline{v'^2}\right) \tag{2}$$

where *u´*, *v´* are the fluctuating in time x and y components of the velocity, respectively. A detailed explanation of the technique can be found in [20].

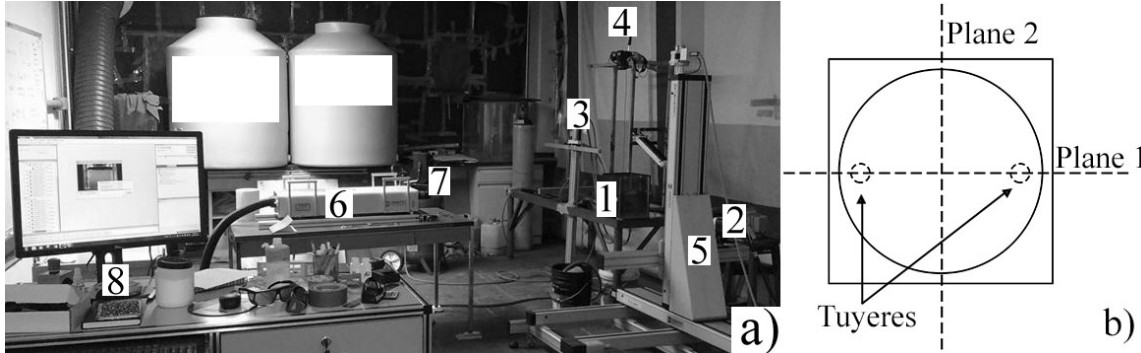

**Figure 1.** (**a**) Experimental arrangement used in the PIV-PLIF (Particle Image Velocimetry-Planar Laser Induced Fluorescence) technique. (1) physical model, (2) high-speed camera, (3) flowmeters, (4) camera, (5) high-speed camera position controllers (isel®Traverse system (isel Germany AG, Eichenzell, Germany)), (6) laser head, (7) air compressor, and (8) computer. (**b**) Measurement planes.

Mixing times were measured using the planar laser-induced fluorescence (PLIF) technique, which has been widely used in mechanical-stirred vessels [24] and has been recently used with success in gas-stirred ladles [7]. The mixing time was measured in the entire Plane 2 (see Figure 1b), using 3 mL of Rhodamine 6G (Sigma-Aldrich, Søborg, Denmark) as the fluorescent tracer, added directly over a gas injection plug. In the case of differentiated flows, the tracer was added on the plug with the higher gas flow rate (75% of the total gas flow rate). Previous to the mixing time experiments, a 20-point in situ calibration was performed using concentrations of Rhodamine 6G ranging from 0.0 up to 4.42 parts per trillion (ppt). The mixing time was computed by tracking the average evolution of the mean concentration of solute in the measurement plane through the PLIF technique. When this mean concentration enters into a window of ±5% of the equilibrium concentration and do not leave thereafter, then, that instant is taken as the mixing time. Mixing time in the physical model ($tmix_{mod}$) is related to the mixing time in the prototype ($tmix_{fs}$) through the dynamic similarity, as presented by Mazumdar et al. [22] in Equation (3), which considers the geometric scale factor ($\lambda$):

$$\frac{tmix_{mod}}{tmix_{fs}} = \lambda^{0.5} \tag{3}$$

The experimental data were processed in MATLAB® R2017b (Mathworks, Natick, MA, USA) with a proper code, obtaining curves of dimensionless concentration versus time and calculating the mixing time over the whole measured Plane 2. Further details of the experimental technique can be found in [25].

## 3. Results and Discussion

As an illustration of the PLIF technique, Figure 2 shows the evolution of Rhodamine 6G injected above the plume of high gas flow rate for experiment G in Table 1. Figure 2a is the rough image at 0.5 s of the tracer injection, while Figure 2b is the concentration contour map obtained from the rough image processing using the calibration of the PLIF technique. Figure 2c shows the rough image for the

same experiment at 4 s and Figure 2d is the processed image to get the concentration contour map at 4 s. When the mean concentration of the whole plane reaches ±5% of the equilibrium concentration, that instant is taken as the mixing time of the experiment.

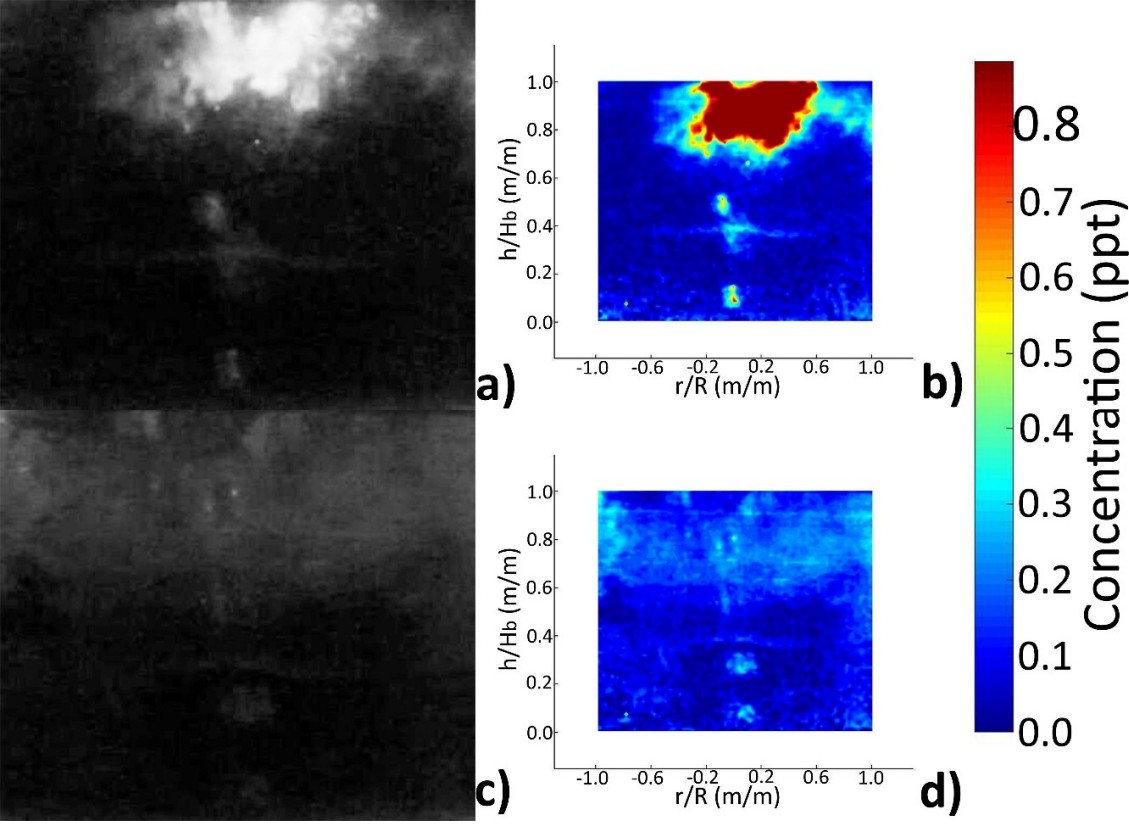

**Figure 2.** Rhodamine 6G evolution in experiment **g** of Table 1. (**a**) Rough image at 0.5 s, (**b**) processed concentration contour map by PLIF (Planar Laser Induced Fluorescence) at 0.5 s, (**c**) rough image at 4 s, and (**d**) processed concentration contour map by PLIF at 4 s.

Figure 3 shows the experimental flow patterns obtained through the PIV technique at different operating conditions. The effect of increasing the gas flow rate was to increase the magnitude of the velocity vectors of the liquid phase. Additionally, an increase in the total gas flow rate, in the case of a 50/50 dual gas injection ratio, increased the size of the circulating loops and decreased the size of low-velocity zones. This equal injection mode showed the well-known upper symmetric toroid circulation loops at the top of the ladle. On the other hand, the use of a non-symmetric dual gas injection ratio of 75/25 (Figure 3c,d,g,h) generated one circulation loop bigger than the other. The smaller recirculation loop, located in the side of the low flow rate input plug, was deformed and dragged toward the center by the larger loop. As a result of this drag effect an increment occurred in the magnitude of the velocity vectors below the smaller recirculation loop, enhancing the mixing and showing, in the measured plane, fewer low-velocity zones than in the conventional blowing mode (equal gas injection ratio). Consequently, it can be expected that at the same gas flow rate and oil thickness, mixing was improved with a differentiated gas injection ratio in comparison with the use of equal flows. An increase in the oil thickness, as observed by comparing Figure 3a–d with Figure 3e–h, respectively, diminished the velocity of the liquid in the vicinity of the oil layer at the top of the ladle. In Figure 3g, the combined effects of the friction exerted by a thick slag and the drag caused by the dual non-symmetric gas injection almost inhibited the smaller loop located to the left. Figure 3h shows the reduction of the bigger circulation loop (right) due to the presence of a thicker oil layer in comparison with Figure 3d (thinner layer of slag), but this bigger loop was smaller than the loop created with the

same slag thickness but lower gas flow rate in Figure 3g, which could be explained by the interaction of both recirculation loops associated with the lower flow rate plug and the thicker oil layer.

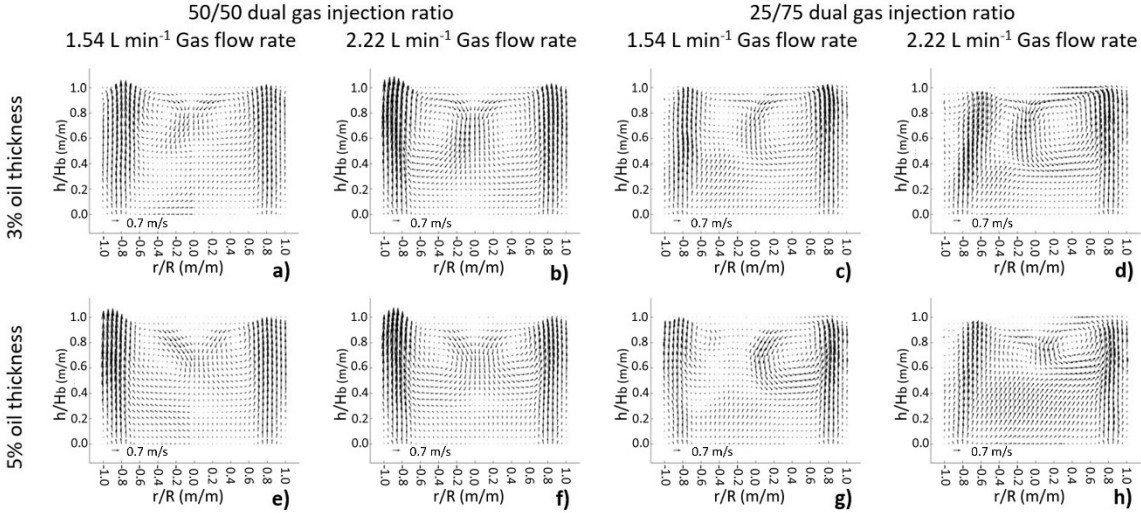

**Figure 3.** Flow patterns of the eight experiments obtained with the PIV (Particle Image Velocimetry) technique in Plane 1. Experiments (**a–h**) (see Table 1 for details).

Figure 4 shows the contours of turbulent kinetic energy (*k*) measured through the PIV technique at different operating conditions. As expected, the movement promoted by the bubbles ascending through the liquid produces high turbulence zones corresponding to the plume zones and the magnitudes of the turbulent kinetic energy in the plane match the water velocity magnitudes (see Figure 3). By increasing the gas flow rate the values of *k* in the circulation loops increase. A thick slag reduces turbulence and the turbulent interaction between the loops. The change from equal (50/50) to differentiated (75/25) dual gas injection promotes more and better distribution of the turbulence; this can be observed clearly by comparing Figure 4b,f with Figure 4d,h, respectively, where the only modified variable was the dual gas injection ratio. The high kinetic energy zones are more evenly distributed throughout the vessel when differentiated injection is used.

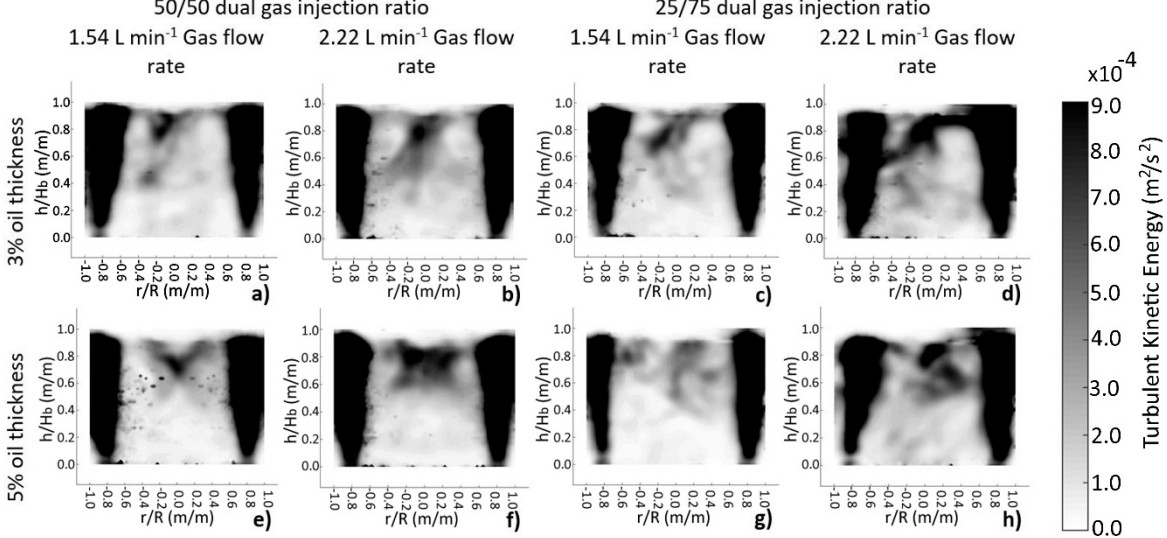

**Figure 4.** Contours of turbulent kinetic energy (*k*) of the eight experiments obtained with the PIV (Particle Image Velocimetry) technique in plane 1. Experiments (**a–h**) (see Table 1 for details).

Results on mixing times of the eight experiments obtained through the PLIF technique [7,25] are presented in Table 2. As expected from the analysis of Figures 3 and 4, the increase of gas flow rate from 1.54 to 2.22 L min$^{-1}$ reduced in all cases the mixing times with the same conditions of slag/oil thickness and dual gas injection ratio. The increment in oil thickness, as expected from the analysis obtained from Figures 3 and 4, increases mixing times, but this increment was less than one second in all cases. The use of a 75/25 dual differentiated flow ratio instead of a 50/50 equal flow resulted in a steeped and surprising decrease in mixing time, as presented in Table 2. The use of differentiated gas injection at the same gas flow rate and oil thickness conditions decreases mixing times between 1 and 2.5 s in all cases. The mixing time obtained with 3% oil thickness, 1.54 L min$^{-1}$ of gas flow rate, and equal gas injection was 5.70 s, but the single change to a differentiated gas injection ratio decreased the mixing time to 3.15 s, which represents a decrease of 2.55 s (45% of reduction). The effect of a 75/25 double injection ratio is less significant at higher gas flow rate (2.22 L min$^{-1}$); with 3% oil thickness the mixing time decreased 1.5 s (37% of reduction) and with 5% oil thickness the reduction in mixing time was around 22%, in comparison with the equal injection under similar operating conditions. All the above-mentioned effects could be explained by the observed changes in experimental flow patterns (Figure 3) and contours of turbulent kinetic energy (Figure 4), where better distribution of the water velocity and turbulence, as well as less low-velocity zones and more high turbulence zones, were achieved with the differentiated double gas injection than with the equal dual injection.

**Table 2.** Averaged mixing times in seconds (s) obtained with the PLIF technique under different operating conditions.

| Oil Thickness (%) | 1.54 L min$^{-1}$ Gas Flow Rate | | 2.22 L min$^{-1}$ Gas Flow Rate | |
|---|---|---|---|---|
| | 50/50 Dual Gas Injection Ratio | 75/25 Dual Gas Injection Ratio | 50/50 Dual Gas Injection Ratio | 75/25 Dual Gas Injection Ratio |
| 3% | 5.70 ± 0.14 | 3.15 ± 0.17 | 4.03 ± 0.31 | 2.53 ± 0.19 |
| 5% | 6.55 ± 1.48 | 4.20 ± 0.29 | 4.13 ± 0.10 | 3.23 ± 0.06 |

Time-averaged slag eye areas are presented for each experiment in Figure 5. In the photographs, almost symmetric slag eyes are generated with a 50/50 dual gas injection ratio, whereas a 75/25 ratio generates big differences in size between both slag eyes. An increase of the oil/slag thickness produces, as expected, slightly smaller slag eyes in all cases. An increase in gas flow rate increases the open eye areas, as can be seen from the comparison of Figure 5a,e with Figure 5b,f, respectively, for equal gas flows, and Figure 5c,g with Figure 5d,h, respectively, for differentiated flows. In order to obtain quantitative results, the experimental slag eye areas were measured with the software MATLAB® R2017b to determine, accurately, the effect of the three analyzed variables on the slag eyes.

The results of the slag eye percentage area of the total surface for the eight experiments are presented in Table 3. As observed in Figure 5, an increase of gas flow rate increases the percentage of slag eye area in all cases, and the contrary effect is obtained with an increase of oil thickness. The impact of the double gas injection ratio was finally determined with the results of Table 3, where a 75/25 ratio produces in all cases bigger slag eyes than those obtained with a 50/50 ratio. These results suggest that a differentiated flow gas injection produces smaller mixing times but at the same time bigger slag eyes than the equal flow injection ratio.

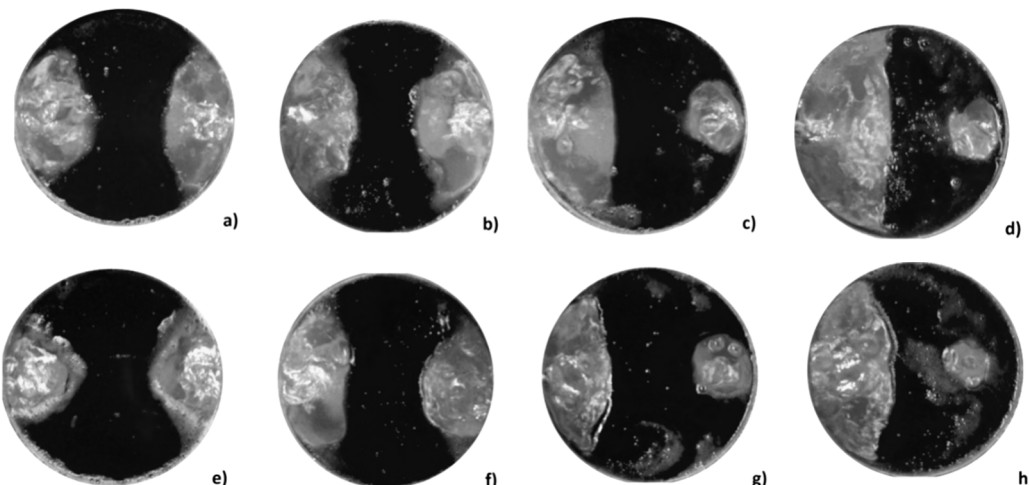

**Figure 5.** Time-averaged photographs of the slag eye at the top of the surface at various operating conditions. Experiments (**a**–**h**) (see Table 1 for details).

**Table 3.** Slag eye area percentage of the total surface area for different operating conditions experimentally obtained through image analysis.

| Oil Thickness (%) | 1.54 L min$^{-1}$ Gas Flow Rate | | 2.22 L min$^{-1}$ Gas Flow Rate | |
|---|---|---|---|---|
| | 50/50 Dual Gas Injection Ratio | 75/25 Dual Gas Injection Ratio | 50/50 Dual Gas Injection Ratio | 75/25 Dual Gas Injection Ratio |
| 3% | 39.40 ± 2.27 | 45.40 ± 3.53 | 51.47 ± 1.49 | 58.35 ± 1.97 |
| 5% | 34.13 ± 1.79 | 34.21 ± 2.96 | 43.99 ± 2.06 | 49.88 ± 3.03 |

With the results of mixing time (*tmix*) shown in Table 2 and the percentage of slag eye area ($A_{eye}$) exposed in Table 3, a statistical analysis was performed to evaluate the effect of the three variables studied in this work (*Q*: gas flow rate, *Ra*: dual gas injection ratio, and *hs*: oil thickness). The results of the main effect of the variables and their interactions over the *tmix* and $A_{eye}$ are presented in Table 4. The variable *Q* had the highest main effect on the $A_{eye}$ with an increase of 12.6 ± 0.5% by increasing the gas flow rate from 1.54 to 2.22 L min$^{-1}$. The second most relevant variable was *hs*, with an adverse effect of 8.1 ± 0.5%, which means that using 5% of slag instead of 3% decreases in more than 8% the area of the open eye. In third place, the variable *Ra* had a positive effect on the $A_{eye}$; the open eye area increases by almost 5% by using differentiated flows. The variable *Ra*, surprisingly, shows the highest main effect on the *tmix*; a 75/25 dual gas injection ratio decreases the *tmix* in 1.70 ± 0.05 s. On the other hand, the variable *Q* decreases 1.30 ± 0.05 s the *tmix*, and *hs* increases it by 0.55 ± 0.05 s. Two of the three two-factor interactions that had significance over the *tmix* involve *Ra*; these are the same double interactions that are significant for $A_{eye}$. The interaction *Q\*Ra* has a mean effect of 1.67 ± 0.50% over $A_{eye}$ and 0.50 ± 0.05 s over *tmix*. This indicates that using differential flows implies a moderate reduction over mixing time and a strong increase in eye area as gas flow rate increases, which suggest that a differentiated flow is more beneficial at intermediate or low gas flow rates. The interaction *Ra\*hs* has a medium effect of −1.73 ± 0.50% over $A_{eye}$ and 0.33 ± 0.05 s over *tmix*, which indicates that the use of differentiated flows causes an increment in mixing time but also a reduction in the $A_{eye}$ as oil thickness increases.

**Table 4.** Calculated effect of the variables and their interactions on slag eye area ($A_{eye}$) and on mixing time (*tmix*).

| Variable | Effect on $A_{eye}$ (%) | Effect on *tmix* (s) |
|---|---|---|
| *Q* (Gas Flow Rate) | $12.64 \pm 0.50$ | $-1.30 \pm 0.05$ |
| *Ra* (Injection Ratio) | $4.71 \pm 0.50$ | $-1.7 \pm 0.05$ |
| *Hs* (Oil Thickness) | $-8.10 \pm 0.50$ | $0.55 \pm 0.05$ |
| *Q\*Ra* | $1.67 \pm 0.50$ | $0.50 \pm 0.05$ |
| *Q\*hs* | $0.13 \pm 0.50$ | $-0.15 \pm 0.05$ |
| *Ra\*hs* | $-1.73 \pm 0.50$ | $0.33 \pm 0.05$ |
| *Q\*Ra\*hs* | $1.24 \pm 0.50$ | $-0.02 \pm 0.05$ |

Equations (4) and (5) were obtained through the statistical analysis to predict the behavior of the $A_{eye}$ and *tmix* in terms of the three variables studied in this work. Both equations were used to perform a multi-objective, multi-variable optimization using the MATLAB® R2017b optimization module and the so-called genetic algorithm (NSGA-II). For a comprehensive review of applications of evolutionary multi-objective optimization, the reader is referred to [26]. The optimization was performed to identify the conditions that simultaneously minimize the mixing time and the open eye area with the purpose of finding the case that presents the best mixing conditions on the plane of measurement and the smallest open eye area in the oil layer. The results of such optimization produced 50 combinations of the three variables, all of them coinciding in the use of differentiated injection with a value of 75/25 dual gas injection ratio, because of the strong main effect of the differential flow and its interactions on mixing time and the open eye area. This result is significant as it strongly states that in dual injection systems the differentiated flow ratio has to be practiced for optimum agitated and slag covered baths.

$$A_{eye} = -47.0 + 41.9\,Q + 118.8\,Ra + 17.0\,hs - 38.5\,Q*Ra - 8.90\,Q*hs - 34.3\,Ra*hs + 14.5\,Q*R*hs \quad (4)$$

$$tmix = 20.31 - 5.20\,Q - 24.6\,Ra - 0.34\,hs + 6.72\,Q*Ra - 0.104\,Q*hs + 1.67\,Ra*hs - 0.20\,Q*Ra*hs \quad (5)$$

The $A_{eye}$ and *tmix* obtained with the 50 optimal combinations were plotted in a denominated Pareto front, presented in Figure 6, where three regimes of behaviors are clearly defined. Firstly, a straight line with a stepped negative slope indicating a significant decrease in $A_{eye}$ with an increase of *tmix* in a narrow range from 2.6 up to 3.3 s (regime 1). Secondly, a linear trend with a low decrease of $A_{eye}$ with *tmix* from 3.3 to 4.2 s (regime 2). Finally, a constant behavior of $A_{eye}$ starting from a *tmix* of 4.2 s (regime 3). To find any pattern across the decision variables for solutions in different zones of the Pareto front, a similarity analysis among Pareto optimal solutions and Pareto characterization was performed. Accordingly, an investigation was carried out in each of the three mixing time regimes identified in Figure 6 to determine whether the obtained *Q-Ra-hs* optimal solutions were showing some kind of similarity in terms of the associated decision variables.

Figure 7 displays the operator´s rules for the three regions showed in the Pareto front from Figure 6. Figure 7a presents the operator's rules for mixing time ranging from 2.6 to 3.3 s, Figure 7b shows those for the range of mixing times between 3.3 and 4.2 s, and Figure 7c shows these rules for mixing times greater than 4.2 s. Although decision variables have specific and discrete operation conditions with their own units, they are shown in a dimensionless relative scale from their lowest (0) to their highest level (1) as presented in Table 1, and they are joined with straight lines to show the trends. In all cases the *Ra* values correspond to the high level explored in this work (differentiated injection ratio), suggesting that for optimal performance the ladle must be operated including differentiated flows. On the other hand, this figure shows that the operational rules regarding *hs* and *Q* change depending on the mixing time interval considered in the analysis. In the first regime (Figure 7a), corresponding to a low mixing time with a high slope in the Pareto front, *hs* must be maintained as low as possible and *Q* can be switched to its high level to enhance agitation or to its low level to keep the area as small as possible. This regime is then controlled by the gas flow rate; if it is increased, it has a strong

effect by reducing the mixing time but also increasing the slag eye area. For intermediate mixing times (Figure 7b), $Q$ must be switched to low values and the slag layer can be adjusted from intermediate to high values to minimize the open eye area maintaining a mixing time as small as possible depending on the desired compromise between the size of the open eye and the desired mixing time. Finally, in the third regime (Figure 7c) for high mixing times, the ladle must be operated using low gas flow rates and high values of slag thickness, with practically the same differentiated injection ratio. It has to be stated for the benefit of the operator that the slag thickness is not expected to be controlled or changed on purpose during ladle operation, but previously selected and specified according to practice, to obtain the expected results in the real industrial procedure and also the range of this variable is narrow in this study, so the results associated with this parameter should be used wisely by the operator

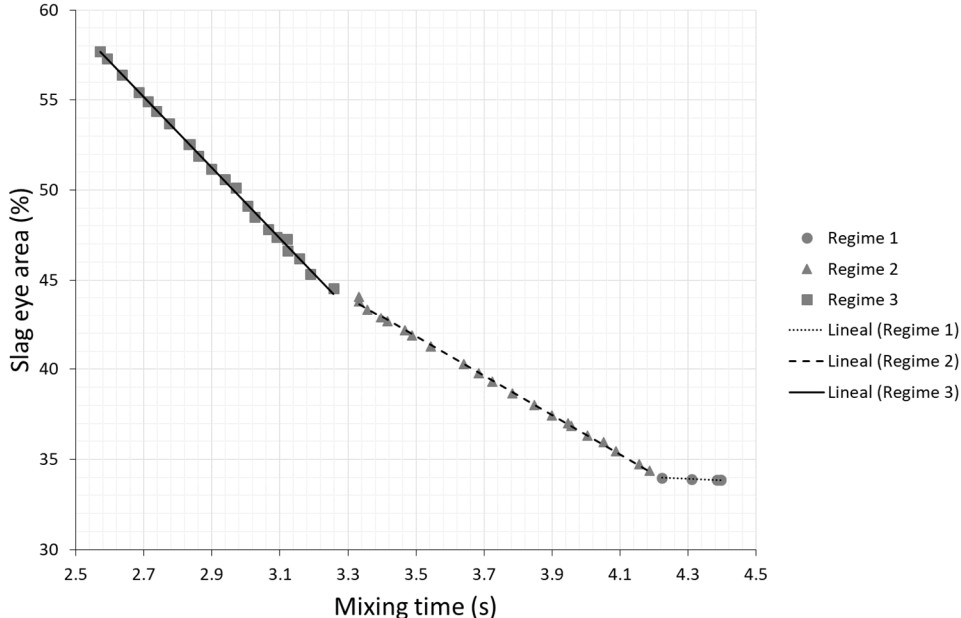

**Figure 6.** Pareto front of the multi-objective optimization of slag eye area ($A_{eye}$) and mixing time (*tmix*).

For practical purposes, in a normal Ladle operation, several operations are performed sequentially and depending on its nature our results could be used for the optimal operation. Desulphurization and deoxidation are earlier operations that are mass controlled and should be accelerated initially by an intense agitation (minimizing mixing time) with an associated big open eye area, and then, after a specific processing time, and in order to avoid the problems associated to such a big eye area, the flowrate can be reduced to intermediate values to keep promoting a good slag metal contact, but reducing the open slag eye area. This reduction in gas flow rate must be performed according to the previous practical experience and results obtained in the specific Steelmaking Plant. The same operation parameters providing intense stirring can be used during the alloying step, which also requires good mixing conditions. Finally, in the last step of refining, dealing with the inclusion removal and thermal homogenization of the bath, it can be used low stirring conditions to enhance the flotation of inclusions and to prevent further oxidation of the steel (minimize eye area) prior to pouring the liquid steel into the caster.

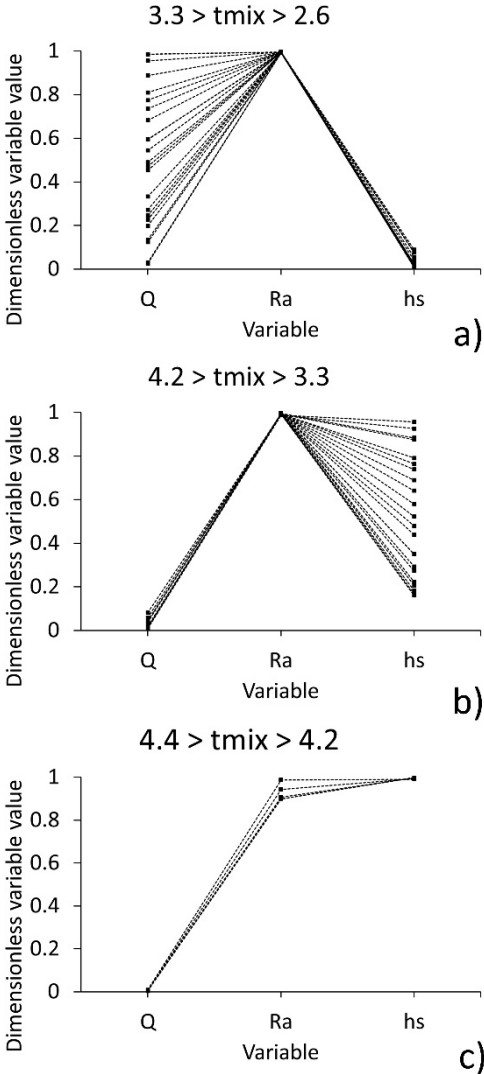

**Figure 7.** Operator´s rules obtained for all variables in dimensionless relative values for the three regimes of mixing times: (**a**) regime 1, (**b**) regime 2, and (**c**) regime 3, shown in the Pareto front (Figure 6).

## 4. Conclusions

The effect of gas flow rate, slag height, and gas blowing mode on mixing time, open eye area, and fluid-dynamics in a gas-stirred ladle was studied using physical modeling, PIV, and PLIF experimental techniques. The main conclusions are:

- The use of differentiated (75/25) dual gas injection ratio instead of equal (50/50) flow rates resulted in a steeped and surprising decrease in mixing time and in a relatively slight increase in open eye area, having a more significant main effect in reducing the mixing time in comparison with the increment in gas flow rate (−1.30 ± 0.05 s for $Q$ versus −1.70 ± 0.05 s for $Ra$), with a moderate increment on the slag eye area (12.6 ± 0.5% for $Q$ versus 4.7 ± 0.5% for $Ra$).

- PIV results showed that the use of a differentiated dual gas injection ratio generates fewer low-velocity zones and more and better distribution of the turbulence compared with the conventional blowing mode. The experimental flow patterns and contours of turbulent kinetic energy confirmed that the use of a differentiated gas injection promotes better mixing conditions compared with equal dual injection.

- The two-factor interactions involving the variable $Ra$ were significant for both mixing time and slag eye area. The interactions with $Q$ implied that differentiated flow is better at low flow rates,

because at high flow rates the decrease in mixing time is reduced and the eye area increment is increased. The interactions with *hs* implied that differentiated flow amplifies the effect of increasing the slag thickness, with a further increase in mixing time and a higher decrease in the eye area.

- The multi-objective optimization resulted in 50 combinations of values of the three variables, divided into three regimes (the first one controlled by the flow rate, the second controlled by the slag height, and the third with fixed operating conditions); all of them presented differentiated gas injection, which suggest that for optimal performance the ladle must be operated using differentiated flow ratios.

- The results of this work can be applied to adjust, in a rational way, the gas flow rates used during operation of the ladle in order to maximize the benefits of using differentiated gas flow rates, as a function of the intensity of the mixing needed for the different processes performed in the ladle.

**Author Contributions:** Conceptualization, M.A.R.-A. and G.T.; methodology, L.E.J.-P.; software, D.R.G.-M. and L.E.J.-P.; formal analysis, C.G.-R.; writing—original draft preparation, L.E.J.-P. and C.G.-R.; writing—review and editing, M.A.R.-A. and G.T.; funding, M.A.R.-A.

**Funding:** This research was funded by DGAPA UNAM, grant number PAPIIT IN115619.

**Acknowledgments:** Luis Enrique Jardón-Pérez, CVU 624968, as a student registered in the Doctoral Program in Chemical Engineering at the Universidad Nacional Autónoma de México (UNAM), thanks CONACyT for financial support through a Ph.D. scholarship. Additionally, authors thank A. Amaro-Villeda for technical assistance through the development of this research work.

**Conflicts of Interest:** The authors declare no conflicts of interest.

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
