# Peer review of "Effect of Differentiated Injection Ratio, Gas Flow Rate, and Slag Thickness on Mixing Time and Open Eye Area in Gas-Stirred Ladle Assisted by Physical Modeling"

_metals, doi:10.3390/met9050555_

Reviewer 1 Report

Authors should explain in the text of paper what means oil thickness expressed in percentage value

Authors should explain in the text of paper how was calculated gas flow rate.

Axis X and Y from figure 2 and 4 are very not readable - too small resolution

Authors should explain in the text of paper how was calculated turbulent kinetic energy from experiments

Authors should explain in the text of paper how was calculated and defined mixing time.

What is relation of calculated mixing time with time for real industrial process.

Author should add to paper figures with tracer concentration for PLIF experiments.

How many times each experiment cases were repeated, please explain in the text.

Author Response

See file attached

Reviewer 2 Report

The paper, generally well structured and supported by an exhaustive preliminary literature analysis, is aimed at investigating the effect of some physical parameters involved in ladle stirring operations on steel mixing times. The main point is the demonstration of the benefit induced by an asymmetrical feeding from the plug. As a matter of fact, slag thickness variation is not expected to be strong in practice, and in Table 1 the range of variation seems to be relatively narrow. For this, an investigation on the effect of the slag chemical-physical properties on eye formation in case of asymmetrical feeding would have been interesting.

The conclusions are consistent with the results. More discussion of the findings would have been appreciated, as in some case there is the feeling of reading a simple list and evidence of results achieved from the experimental work (e.g., discussion of the Pareto combined line of figure 5). 

Concerning the modelling apparatus, details on the physical model scaling, at least for what concerns the criterion of similarity the set up of the model are missing. Also considerations on the possible wall effects in a so small (1:17) reduced scale model can be expected. 

Thanks to the Authors for their contribution.

Author Response

See file attached

Reviewer 3 Report

Dear Authors,

your paper Effect of Differentiated Injection Ratio, Gas Flow Rate, and Slag Thickness on Mixing Time and Open Eye area in Gas-Stirred Ladle Assisted by Physical Modeling presents a modelling of a LF process with the attempt to optimize process parameters in order to reduce mixing time (i.e., the overall time of LF treatment) and the open-eye (harmful for steel quality).

The introduction is really exhaustive with an in-dept literature review that clarify the position of your paper with respect the state-of-the-art.

Experimental procedure needs some minor clarification

Results section is well organized and present the result in clear and simply way

Conclusions should be improvement with some practical comments

Please find in the following some comments/remarks that I would like to pose to the Authors attention:

a) Please, in the introduction, to avoid mistaking with the gas injection ratio (R) use "r" to define the position of the porous plug and specify at line 41 that "r" is referred to the radius of the ladle.

b) in a paper of 2010 by Geng (ISIJ Vol. 50, issue 11, pp. 1597-1605) it was shown that the best plug distribution on the LF bottom is three plug at r/2, two on the same plane, and one perpendicular to the previous two. With this configuration the steel motion follows exactly the theoretical motion that would that the steels (and thus the inclusions) moves fast from bottom to top of the ladle and then move slow horizontally under the top slag. In your introduction, all the models and papers cited are referring only with a two plugs configuration. Could you add some comments regard this?

b) about experimental procedure, have the Authors check if the density ratio between water and oil, as well as the viscosity ratio are comparable with those of steel and slag? Have the Authors also check if the use of air could well replicate the effect of Ar in a steel bath? these two aspect are fundamental to validate the design of experiment

c) have the Authors chose the cylinder size according to the Buckingham π theorem? I mean, is the reactor design representative of the fluid-dynamic phenomena occurring in a ladle?

d) could the Authors specify the criterion used to estimate the mixing time? Is the time after that all the tracer is dissolved or is equally distributed?

e) Table 2 and 3: the standard deviation shown in the results is calculated from the several measurement of PIV and PLIF or is the standard deviation based on the two replicas?

f) in conclusions, could the Authors add a summarized and general indication about the best parameters should be used to both minimize the time and reducing the open-eye (I know it is process time depending)

g) how it is possible to link your results with the common practice (as described by Turkdogan in

the making, shaping and treating of steel) to have high stirring during the de-S and de-Ox addition and low stirring to maximize the slag interaction with the de-S and de-Ox products? Typically, a suggestion given to LF operator is to work with 1/3 of the total process time with high Ar rate (non consecutive) and for the 2/3 with low Ar rate.

Author Response

See file attached

Round  2

Reviewer 3 Report

Dear Authors,

thank you for your kind reply

I have no more comments to pose to your attention

Best Regards